# The Influence of Cellulose Ethers on the Physico-Chemical Properties, Structure and Lipid Digestibility of Animal Fat Emulsions Stabilized by Soy Protein

**DOI:** 10.3390/foods11050738

**Published:** 2022-03-02

**Authors:** Susana Cofrades, Arancha Saiz, Miriam Pérez-Mateos, Alba Garcimartín, Rocío Redondo-Castillejo, Aranzazu Bocanegra, Juana Benedí, María Dolores Álvarez

**Affiliations:** 1Institute of Food Science, Technology and Nutrition (ICTAN-CSIC), 28040 Madrid, Spain; a.saiz@ictan.csic.es (A.S.); miriam@ictan.csic.es (M.P.-M.); mayoyes@ictan.csic.es (M.D.Á.); 2Pharmacology, Pharmacognosy and Botany Department, Pharmacy School, Complutense University of Madrid, 28040 Madrid, Spain; roredond@ucm.es (R.R.-C.); aranboca@ucm.es (A.B.); jbenedi@ucm.es (J.B.)

**Keywords:** emulsions, cellulose ethers, rheological properties, microstructure, lipid digestibility

## Abstract

This study explores the influence of carboxymethylcelullose (CMC) and methylcelullose (MC), added by simultaneous (sim) and sequential (seq) emulsification methods, on the structure, rheological parameters and in vitro lipid digestibility of pork lard O/W emulsions stabilized by soy protein concentrate (SPC). Five emulsions (SPC, SPC/CMC-sim, SPC/CMC-seq SPC/MC-sim, SPC/MC-seq) were prepared in vitro. The presence of CMC and MC, and the stage of incorporation affected the emulsion microstructure. In the SPC emulsion, lipid droplets were entrapped by a protein layer that was thicker when MC was added, providing greater resistance against environmental stresses during gastrointestinal digestion. At 37 °C, CMC incorporation produced a structural reinforcement of the SPC emulsion, whereas MC addition did not affect the network rigidity, although a delaying effect on the crossover temperature was observed, which was more evident in SPC/MC–seq. The presence and stage of CMC and MC incorporation affected the rate and extent of lipolysis, with SPC/MC-seq presenting an inferior concentration of free fatty acids. The lower extent of lipolysis observed in SPC/MC-seq may be positive in the manufacture of animal fat products in which reduced fatty acid absorption is intended.

## 1. Introduction

Animal fat is a functional ingredient that provides quality and acceptable sensory properties mainly in meat products and several bakery products. However, animal fat contains a relatively high amount of saturated fatty acids (SFA) associated with a high risk of cardiovascular disease, diabetes, and obesity [1]. Therefore, consumers are reducing the common intake of this type of product. To improve the fat content (amount and fatty acid profile) of several meat derivatives, various technological options have been used in the last decades, which include the replacement of animal fat with vegetable or marine oils or the incorporation of fat substitutes [2,3]. With these options, however, the quality characteristics, and mainly the sensory attributes of the reformulated products, are negatively affected in contrast with their whole animal fat meat counterparts. In this context, a new strategy to achieve healthy meat derivates without altering their organoleptic quality could be attained by reducing the lipid digestibility of the whole animal fat meat products. For this purpose, a first step would be to design pork fat emulsions with controlled lipid digestibility for their subsequent incorporation in the reformulation of emulsified meat products, such as pâté, frankfurter type sausages, etc.

Polysaccharides and proteins are important functional ingredients that often co-exist in food emulsions. Proteins have amphiphilic nature and exhibit surface activity, playing a main role in the stabilization and formation of emulsions by decreasing the interfacial tension and by making a protective membrane around oil droplets [4,5]. Particularly, soy proteins (isolated and concentrate) are important food ingredients due to their good functional (emulsifying/foaming) and nutritional properties, as well as for their cholesterol-lowering effect [6]. 

Most high molecular weight polysaccharides, being hydrophilic, are not prone to be adsorbed at the interface of emulsions. Nevertheless, they can act as emulsion stabilizers by increasing the viscosity or by gelling the aqueous phase, as well as by inducing the flocculation of emulsion droplets through bridging or depletion mechanisms, depending on the adsorbing properties of the polysaccharide [7,8]. Among polysaccharides, cellulose ethers have a strong tendency to accumulate at the oil/water interface [9] due to their high surface activity, which confers them to numerous technological applications in the food industry [8]. Depending on their chemical structure, cellulose ethers show different surface activities. The higher the degree of total substitution, the higher the hydrophobicity of the polysaccharide, and thus, the increase in surface/interfacial activity [10,11]. Carboxymethylcellulose is one of the most accessible cellulose ethers and has been utilized in different food applications because it ionizes in water [12]. Like many other anionic polysaccharides, CMC can tune the charge density and surface adsorption capacity of the hydrocolloids [13], which may contribute to their emulsion abilities. 

It is widely known that mixed protein–polysaccharide systems are well-recognized macromolecular assemblies capable of stabilizing oil-in-water (O/W) food emulsions [14]. In this context, the presence of polysaccharides can significantly improve protein functionality, for instance, in terms of emulsifying properties, although this depends on the extent of their interactions and mass ratio, local environmental conditions, and external inputs such as temperature and shear. In this sense, in the formulation and stabilization of O/W emulsions, oil–water interfacial adsorption of protein–polysaccharide complexes critically depends on whether complexation occurs before, during, or after emulsification, as the order of addition generates different structures and/or compositions at the interface [14,15].

Up to date, the use of CMC has been shown to improve the emulsion properties of many proteins and others biopolymers, such as chitin, microcrystalline cellulose, etc. [14,15,16,17]. Most of these studies have been mainly aimed at improving the rheological properties and stability of low-fat emulsions, as well as the influence of the stage of incorporation of the hydrocolloid in the emulsion (simultaneous or sequential adsorption methods or principles) [15]. Thus, different authors have studied the impact of adding CMC to emulsions stabilized by different proteins, such as whey protein (WP) [15], okara protein [16], and sodium caseinate [18] on the physico-chemical properties of the cited emulsions. In turn, methylcellulose is a non-ionic polysaccharide that exhibits higher hydrophobicity and, consequently, higher surface activity than anionic polysaccharides like CMC. However, there are hardly any studies on the effect of MC on the properties of protein-stabilized emulsions. Reiffers-Magnani et al. [19] studied the stability of WP-stabilized emulsions containing MC added after emulsification in their bulk phase. 

On the other hand, only a few publications have addressed the impact of polysaccharides on the fat digestibility of emulsions stabilized by proteins during their in vitro digestion, and the existing studies have been carried out with vegetable oils (sunflower, soy, etc.). For instance, Malinauskytė et al. [20] analyzed the effect of CMC on in vitro the lipid digestion of WP-stabilized emulsions made with rapeseed oil. In turn, Bellesi et al. [21] studied the lipolysis of soy protein isolated (SPI)-stabilized emulsions containing hydroxypropyl methylcellulose (HPMC) elaborated with sunflower oil under in vitro physiological conditions. The results showed that the extent and rate of lipolysis of emulsions made with plant oils and stabilized by mixtures of proteins and polysaccharides depend on the composition, structure, charge of the interfacial layers and thickness, which determine how the oil drops of the emulsion interact with each other and how they will be digested. 

Furthermore, no research has been found on the effect of hydrocolloid addition on lipid digestion of protein-stabilized pork fat emulsions. Santiaguín-Padilla et al. [22] and Hur et al. [23] studied the effect of encapsulating animal fat in emulsion-type sausages with different biopolymers (pectin, chitosan, and cellulose), albeit in the absence of protein, and found that the lipid digestibility of these sausages was reduced after in vitro gastrointestinal digestion. 

Accordingly, the objective of this research was to determine the role of two different cellulose ethers (anionic CMC and non-ionic MC) on the physico-chemical and structural properties, as well as on the lipid digestibility of pork lard O/W emulsions stabilized by soy protein concentrate. Also, both CMC and MC were added at two different stages of the emulsion formulation, that is, the polysaccharides were added to the emulsions that had been stabilized by SPC using simultaneous/co-adsorption and sequential methods of addition.

## 2. Materials and Methods

### 2.1. Materials and Reagents 

Soy protein concentrate (SPC) (Arcon^®^ S; ADM) was kindly donated by Lactotecnia S.L. Ingredientes Alimentarios (Barcelona, Spain). As stated by the supplier, SPC contains a minimum of 72.0% protein. Carboxymethylcellulose (WALOCEL CRT 30000; The Down Chemical Company, Germany), with a viscosity of 30,000 mPa s in 2% solution at 25 °C, and methylcellulose (METHOCEL A4M, The Down Chemical Company, Germany), with a viscosity of 4000 mPa s at 2% in water at 25 °C, were kindly donated by Brenntag Food & Nutrition (Barcelona, Spain). Pork lard was obtained after a process of clarification of Iberian pork fat, containing 99.9% fat. The enzymes used, including pepsin from porcine gastric mucose (≥2500 U/mg, P7012), pancreatin from porcine pancreas (8xUSP, P7545) and bile extract porcine (B8631) were acquired from Sigma Aldrich Chemie GmbH (Steinheim, Germany). Fluorophores, Red Nile (72485) and Fast Green (F7252) were also from Sigma Aldrich Chemie GmbH (Steinheim, Germany).

Sodium chloride, hydrochloric acid 37%, and sodium hydroxide pellets were purchased from Panreac (Barcelona, Spain). Ethanol (96%) was obtained from CIDAS (Murcia, Spain). Sodium hydrogen carbonate, sodium bicarbonate, potassium chloride, and dihydrate calcium chloride were acquired from Sigma Aldrich Chemie GmbH (Steinheim, Germany). Magnesium chloride hexahydrate, molecular biology grade was purchased from EMD Millipore Corp. (Billerica, MA, USA).

### 2.2. Preparation of Stock Solutions and Emulsions

First, one stock solution containing 9.67% SPC and 0.02% sodium azide and two other stock solutions containing 3.45% CMC or MC were prepared. The solutions were stirred for at least 2 h and then stored for 72 h to ensure complete hydration. 

Then, five different O/W emulsions were prepared by simultaneous and sequential methods at pH 7 as previously described [15]. Pork lard was heated at 60 °C in a water bath until fully melted.

By using the simultaneous or co-adsorption method, the appropriate amounts of CMC or MC and SPC solutions were mixed for 5–6 min at 20,000 rpm using an Ultraturrax (IKA-25, Staufen, Germany), the pH was adjusted to 7 (with 1 M HCl) and the mixtures were stored for 1 h, forming SPC-CMC or SPC-MC complexes. After 1 h, each mixture was homogenized with the melted pork lard for 5 min at 25,000 rpm and stored for 24 h at 4 °C before use. These emulsions were coded as SPC/CMC–sim and SPC/MC–sim, respectively.

At the same time, using the sequential adsorption method, the appropriate amounts of SPC solution and melted pork fat were homogenized for 2 min at 13,200 rpm using the same homogenizer above mentioned to create a primary emulsion. After that, the pH was adjusted to 7 (with 1M HCl) and the emulsion was stored for 1 h to allow the protein to coat fat globules. Thereafter, this primary emulsion was homogenized for 5–6 min at 20,000 rpm with the appropriate amounts of CMC or MC solutions and stored for 24 h at 4 °C before use. These emulsions were coded as SPC/CMC–seq and SPC/MC–seq, respectively.

The final emulsions contained 40% fat, 3% SPC, and 1% CMC or MC. The control emulsion (SPC) contained 40% fat and 4% SPC. The final weight of each emulsion was 200 g.

### 2.3. Fatty Acid Profile of Pork Lard and Emulsions 

Fatty acid contents were determined (in triplicate) in pork lard and emulsions previously freeze-dried as described Gómez-Estaca et al. [24]. Fatty acids were identified by comparison of retention times with a fatty acid standard (Supelco 37 FAME Mix 47885-U, Bellefonte, PA, USA) and the internal standard C13:0 was used for quantification, which was added to the sample in the non-methylated state before methylation. Results were expressed as mg fatty acid/g sample.

### 2.4. Structural Properties

#### 2.4.1. Measurement of Droplet Size and Size Distribution

The particle size and distribution of the emulsions were determined using a particle size analyzer (Malvern Mastersizer S, Malvern Instrument Ltd., Worcestershire, UK) and Malvern Instruments software (v2.19), equipped with a He-Ne laser (λ = 633 nm) and a measurement range of 0.5–900 μm. The emulsions were diluted about 10-fold with distilled water. Then, an aliquot was added to the sample dispersion unit until the obscuration was within the range of 8–15%. Particle size calculations were based on the Mie Scattering theory. At least three measurements were performed on each emulsion. Droplet size was registered as the volume-mean diameter D[4,3] and the equivalent volume-surface mean diameter D[3,2].

#### 2.4.2. Confocal Laser Scanning Microscopy

Fluorophores were added to a freshly made emulsion drop (≈10 µL) over an optical glass. First, 5 µL of Fast Green (0.01% in water) was used as a protein and hydrocolloid fluorophore in the red region spectrum (showed in red color), with laser line excitation at 633 nm and emission range of 650–720 nm. Then, 5 µL of Red Nile (0.01% in ethanol) was used as a lipid fluorophore in the green region spectrum (showed in green color), with laser line excitation at 488 nm and emission range of 505–560 nm. Images were recorded with a confocal microscope (Leica TCS SP5 AOBS, Mannheim, Germany) with 20 × optics, using an optical zoom of 5× in selected areas. Laser lines were provided by an Ar laser and a DPSS laser. Detection ranges were set to eliminate crosstalk between fluorophores.

### 2.5. Rheological Measurements

Rheological measurements of emulsions were carried out with a rotational Kinexus pro rheometer (Malvern Instruments Ltd., Worcestershire, UK) equipped with rSpace software and a high-temperature cartridge in the lower plate for temperature control (resolution to 0.01 °C). Dynamic viscoelastic properties were obtained at both 5 and 37 °C by using a plate-plate measuring geometry PUS40:PLS61X, with a 40 mm diameter serrated upper plate and a 61 mm serrated lower plate (1.0-mm gap), whereas steady shear rheological measurements were made at 37 °C by using the same plate-plate geometry. A cover cell was used to maintain the samples at the specified temperatures and prevent evaporation. All measurements were repeated at least three times and performed 24 h after the formulation of the different emulsions.

#### 2.5.1. Dynamic Viscoelastic Properties

First, to determine the extent of the linear viscoelastic (LVE) range at 5 and 37 °C, stress amplitude sweeps were conducted at 1 Hz varying the stress from 0.002 to 200 Pa, depending on the consistency of the emulsion and the measurement temperature. Then, frequency sweeps were performed at both measurement temperatures between 10 and 0.1 Hz at a chosen stress within the LVE range, 10 points per decade, and rheological properties at a frequency of 1 Hz were derived from the mechanical spectra. Additionally, to analyze the effect of heating on the emulsion structure, temperature sweeps were carried out from 5 to 60 °C at a linear heating rate of 5 °C/min, and at a frequency of 1 Hz. For this, the imposed shear stresses were chosen within the LVE range at 5 °C. However, below 60 °C, a phase change occurred in some of the emulsions formulated and, consequently, the applied stresses do not guarantee the LVE condition of all the samples during heating. From all the different tests performed, storage modulus (*G*′, Pa), loss modulus (*G*″, Pa), complex modulus (*G**, Pa) and loss tangent (tan *δ* = *G*″/*G*′, dimensionless) values were recorded.

#### 2.5.2. Steady Shear Rheological Measurements

To analyze the viscous flow behavior, flow curves were obtained as a function of shear rate ranging from 100 to 0.1 s^−1^ with 10 samples per decade. Values of shear rate, apparent viscosity, and shear stress were recorded. Data from the flow curves of fresh emulsions were fitted to both the power law model. From the power law model ($\eta_{a}\left(\dot{\gamma }\right)=K{\dot{\gamma }}^{n-1}$) and the apparent viscosity ($\eta_{a}$, Pa s) data, the consistency coefficient (K, Pa s^n^) and the flow behavior index (*n*, dimensionless) were derived. In the power law model, $\dot{\gamma }$ is the shear rate (s^–1^).

In addition, a three-step shear rate test was carried out for viscometry rebuild analysis. In the first stage, the samples were subjected to a shear rate of 0.1 s^−1^ for 30 s. Then, in the second stage, a shear rate of 100 s^−1^ was applied for 30 s to cause the breakdown of the emulsion’s structure. Finally, in the third stage, the shear rate was again reduced to 0.1 s^−1^, monitoring the viscosity recovery for 360 s. To compare the time-dependent flow behavior between emulsions, the percentage of viscosity recovery at the end of the test was considered as a relative measure of the time dependence of the apparent viscosity of each emulsion [25].

### 2.6. Texture Measurements

A TA.HDPlus Texture Analyzer (Stable Micro Systems, Ltd., Godalming, UK) provided with Texture Exponent software (version 6.1.16.0) and equipped with a 5 kg load cell was used for the emulsion penetration tests. The tests were carried out at 5 °C using a polyoxymethylene cylindrical probe (P/10, 10 mm Ø) to penetrate each sample to a depth of 10 mm at a rate of 0.50 mm/s. The hardness of the samples was recorded as the maximum force (N) and the area (N s) under the force-time curve at the defined penetration distance. Measurements were performed at least in triplicate and carried out 24 h after formulation.

A penetration test was performed by using a TA.HDPlus Texture Analyzer (Stable Micro Systems, Ltd., Godalming, UK) equipped with a 50 N load cell. A cylindrical probe (P/10, 10 mm Ø) was used to penetrate each emulsion at a rate of 0.50 mm/s up to a depth of 10 mm. The parameters reported were the maximum force (N) and the area (N s) under the force-time curve. At least three measurements per emulsion were carried out 24 h after formulation, which were analyzed with Texture Exponent software (version 6.1.16.0).

### 2.7. In Vitro Digestion

After one day of storage at 4 °C, the different types of emulsions were subjected to a complete simulated in vitro INFOGEST 2.0 digestion method (oral, gastric, and intestinal phases), following the recommendations and fluid composition described by Brodkorb et al. [26] with brief modifications. In order to focus on lipid digestion in the intestinal phase (where most lipid digestion occurs), α-amylase was omitted from the oral phase and gastric lipase was not included. In turn, in the simulated fluids, NaHCO_3_ was replaced by NaCl at the same molarity in order to prevent the pH from rising above 7. The temperature was maintained at 37 °C under continuous stirring during the whole digestion process, and pre-warmed solutions were used throughout the procedure to avoid any temperature fluctuations (which could affect enzyme activity). A pre-test was performed using a pH-stat (TitroMatic 1S, Crison, Alella, Spain) to fix the NaOH and HCl amounts, and then three replicates were done in a shaking incubator (311DS Labnet, Edison, NJ, USA) under the same conditions.

Briefly:

Oral phase: 5 g of each emulsion (containing 2 g of fat) was mixed with 4 mL of simulated salivary fluid (SSF) with no salivary α-amylase (no starch in the emulsion), 0.025 mL CaCl_2_, and 0.975 mL of water. The bolus was incubated at 37 °C for 2 min under continuous stirring using a mechanical shaking device.

Gastric phase: 8 mL of simulated gastric fluid (SGF) electrolyte solution was added, followed by 0.005 of CaCl_2_, and the pH was adjusted to 3 using HCl 1 N. Then, freshly prepared porcine pepsin (0.5 mL of water) was added to obtain 2000 U/mL in the final digestion mixture. The mixture was incubated at 37 °C for 2 h under continuous agitation using the above-mentioned mechanical shaking device. During the two hours of gastric digestion, the pH was adjusted to 3 as required. Finally, the necessary amount of water was added to obtain 10 mL of SGF.

Intestinal phase: 7 mL of electrolyte simulated intestinal fluid (SIF) and 0.04 mL CaCl_2_ were added to the chyme, and the pH was adjusted to 7.0 with NaOH 5 N or 0.1 N under continuous agitation at 37 °C. A solution of bile extract in 4 mL SIF (10 mM in the final mixture) was then added and, if necessary, the pH was readjusted to 7.0. Thereafter, freshly prepared pancreatin in 5 mL of SIF was added with pancreatic lipase (100 U/mL trypsin activity and 2000 U/mL lipase activity in the final mixture). An automatic titration device (TitroMatic 1S, Crison, Alella, Spain) was used to maintain the sample at pH 7.0 by adding NaOH (1 M). The sample was kept under simulated small intestinal conditions for 90 min. To inhibit lipolysis, 200 µL of 5 mM 4-bromophenylboronic acid were added to the final digested mixture and stored at −20 °C until used.

### 2.8. Extent of Lipolysis during In Vitro Digestion

The lipid composition was determined in the emulsion samples and in the pork lard before in vitro digestion, 30 min after the intestinal phase and at the end of the digestion (90 min after the intestinal phase). In all cases, the same amount of fat was used for in vitro digestions for better comparison.

#### 2.8.1. Fat Extraction

Briefly, samples were mixed with chloroform/methanol (1-/1, *v*/*v*) then washed again with chloroform. The organic phase was finally purified using a chloroform/methanol/0.58% NaCl solution mix (*v*/*v*/*v*, 3/48/47) and dehydrated by filtration through anhydrous sodium sulfate. The solvent was evaporated to dryness in a water bath at 40–50 °C under a nitrogen atmosphere.

#### 2.8.2. High-Performance Size-Exclusion Liquid Chromatography (HPSEC)

HPSEC was performed to elucidate lipid composition as described by Dobarganes et al. [27]. Triacylglycerides (TAG), diacylglycerides (DAG), monoacylglycerides (MAG), and free fatty acids (FFA) were quantified at baseline and in the intestinal phase (t = 30 or 90 min). Digestibility of samples was also calculated according to the following formula:$$\mathrm{Digestibility}\;\left(\%\right)=\frac{\left({\mathrm{TAG}}_{\mathrm{initial}}-{\mathrm{TAG}}_{t}\right)}{{\mathrm{TAG}}_{\mathrm{initial}}}\times 100$$

### 2.9. Statistical Analysis

For the fatty acid profile, the droplet size, the rheological and textural measurements of the emulsions, as well as the lipid hydrolysis values during the different stages (30 and 90 min) of the simulated in vitro digestion, a one-way analysis of variance (ANOVA) was performed to compare the different formulated emulsions. Significant differences between pairs of means were evaluated by the Tukey test, using a 95% confidence interval (*p* < 0.05). Analyses were performed using IBM SPSS for Windows, Version 26.0 (IBM Corp., Armonk, NY, USA).

## 3. Results and Discussion

### 3.1. Fatty Acid Profile of Pork Lard and Emulsions

The lipid composition of the pork lard and of the 40/60 O/W emulsions formulated is presented in Table 1. As expected, the level of fat in the emulsions was approximately 60% lower than in the pork lard, in line with the emulsion formulation. As can be seen, in both types of samples, the main fatty acids are MUFA, mostly oleic acid, which implies that both lard and the different emulsions formulated with this animal fat should not be considered as unhealthy as it is commonly done. However, the SFA content is also high. Thus, reducing the lipid digestibility of these emulsions, which will subsequently be incorporated into different meat products, would help enormously to make them healthier, in accordance with current nutritional recommendations.

### 3.2. General Appearance and Structural Properties

The droplet morphology of the emulsions determined by confocal microscopy is shown in Figure 1. In the inset, the general appearance and droplet size distributions determined by light scattering of the fresh emulsions initially stabilized by SPC, and then by the incorporation of two cellulose ethers (CMC and MC) at two different stages of the emulsification process (simultaneous and sequential adsorption methods), can be observed.

In reference to the visual appearance, emulsion SPC presented a less homogeneous, grainy aspect with little consistency, whereas the additional presence of CMC and MC, regardless of the stage of incorporation, resulted in emulsions with a more homogeneous appearance and gel-like characteristics, indicating that both cellulose ethers stabilize the emulsion structure. The sequential addition of CMC and MC (Figure 1c,e) resulted in visually more consistent and cohesive gels than those formed by the simultaneous incorporation (Figure 1b,d). This result is consistent with the fact that mixed protein–polysaccharide systems are capable of stabilizing O/W food emulsions better than proteins alone [14].

On the other hand, all emulsions presented a range of different lipid droplet sizes (green regions) dispersed in an aqueous continuous phase (dark regions). The microstructure showed that the oil droplets of the SPC-stabilized emulsion (in absence of any other hydrocolloid) were homogeneously spread throughout the aqueous phase and exhibited a monomodal but broad distribution with a predominant population at around 55 μm (Figure 1a), corroborating previous findings [14]. This emulsion was expected to be negatively charged, as it is well known that emulsions stabilized by soy protein at pH 7 (well above the isoeletric point PI ≈ 4.5) present a high negative charge [28,29]. The presence of both cellulose ethers (CMC or MC) in the emulsions, as well as the stage of incorporation (simultaneous or sequential), seemed to have a significant influence on the microstructure and therefore on the particle size distributions, as reflected by the oil droplets, which were mostly more spherical, smaller and more homogeneous than those of the emulsion stabilized by SPC alone (Figure 1a).

Regardless of the adsorption method (simultaneous or sequential), emulsions SPC/CMC (Figure 1b,c) presented a more compact structure in comparison with emulsions SPC/MC (Figure 1d,e), probably due to the higher charge density of CMC (anionic cellulose ether) compared with MC (non-ionic). In emulsions SPC/CMC-sim and in SPC/CMC-seq (Figure 1b,c), some flocks of oil droplets can be observed, suggesting that lipid droplets are held together by relatively weak attractive forces. This behavior is characteristic of a depletion flocculation mechanism, which is known to occur when relatively high concentrations of non-adsorbed anionic hydrocolloid molecules are present in the aqueous phase surrounding the lipid droplets [5,27]. Indeed, it has been previously reported that the addition of 0.1% fucoidans or higher can promote depletion flocculation in caseinate-stabilized emulsions at pH values above 6, where the droplets and the biopolymer are both strongly negatively charged, and consequently, there is a strong electrostatic repulsion between them [30]. However, this mechanism was not detected as much in the SPC-stabilized emulsion in which MC was incorporated simultaneously or sequentially (Figure 1d,e), probably because on the one hand, MC does not present any charge, and on the other, as it is more hydrophobic than CMC, it presents more affinity for the oil droplet interface, giving rise to a thicker layer, which stabilizes the emulsion without flocculation or coalescence phenomena. In fact, in Figure 2, it can also be observed that the lipid droplets (black) were trapped by a thin protein layer (red) in emulsion SPC (Figure 2a), and it is possible to appreciate that this layer was thicker when the cellulose ethers were incorporated, mainly in the emulsion containing MC (Figure 2b,c), which could supply higher resistance against environmental stresses such as changes in ionic strength or in pH occurring during the gastrointestinal digestion. In this sense, Bellesi et al. [21] reported the importance of the interfacial films in the mixed protein-polysaccharide emulsions, as they can affect the oil droplet size, thus altering their behavior during gastrointestinal digestion.

Table 2 shows D[4,3] and D[3,2] values related with destabilization processes [21]. Overall, the size of the oil droplets was larger in emulsion SPC, as observed in Figure 1a, reflecting that SPC alone could be less effective than SPC/CMC and SPC/MC complexes in producing small droplets during emulsification. The conformational flexibility of soy proteins is also poorer than that of animal proteins, thus causing a relatively less capacity to stabilize emulsions [31]. Meanwhile, mixed protein–polysaccharide systems are well-recognized for their ability to stabilize oil-in-water food emulsions [14] enhancing the emulsifying properties of soy proteins as a result of the decrease in interfacial tension [32]. Also, the order of addition of biopolymers generates different structures and/or compositions at the interface [33], as reflected by the different particle size distributions shown in Figure 1. Indeed, in the mixed systems, an increase in the specific surface area (lower D[4,3] and D[3,2] values) was observed when the biopolymers were added by the sequential method (Table 1), associated with a second emulsification step after SPC adsorption at the O/W interface, as described in Section 2.2.

### 3.3. Dynamic Viscoelastic Properties

#### 3.3.1. Stress Sweep Tests

First, the LVE range of the five emulsions at both 5 and 37 °C was determined. The variation of complex modulus (*G**) values versus the applied shear stress wave amplitude is shown in Appendix A. At 5 °C, the control SPC showed the highest *G** values, and therefore, the most rigid fat-crystal network, while the *G** values of both emulsions containing MC were higher than those of the emulsions with added CMC (Appendix A). However, the extent of the LVE range was quite similar for all the different emulsions at the lower temperature, with the viscoelastic limit ranging from approximately 10 to 70 Pa. In turn, at 37 °C, emulsion SPC/CMC–sim exhibited the highest rigidity, followed by SPC/CMC–seq, SPC/MC–seq, SPC and SPC/MC–sim, in that order (Appendix A). Additionally, SPC exhibited a much narrower LVE range, with the lowest density and conformational flexibility (ability to undergo conformational rearrangement) [34]. Still, although there were very small differences between the *G** values of emulsions SPC, SPC/MC–sim and SPC/MC–seq at 37 °C, SPC/MC–seq exhibited a noticeably wider LVE range and therefore higher structural stability than either SPC or SPC/MC–sim. Thus, at 37 °C, the subsequent addition of MC into the SPC-coated oil droplets resulted in a larger LVE region without affecting the network rigidity in comparison with the control emulsion SPC.

#### 3.3.2. Frequency Sweep Tests

The mechanical spectra of the emulsions at 5 and 37 °C are shown in Figure 3. At both temperatures, all emulsions presented a higher storage modulus (*G*′) than the loss modulus (*G*″), and in the frequency range studied, both moduli showed a considerable frequency dependence, therefore exhibiting a weak gel-like or structured liquid behavior as reported by Nishinari [35]. However, at 5 °C, samples SPC and SPC/MC–sim (Figure 3a) exhibited the highest modulus values and the lowest frequency dependence. In turn, at 37 °C, all four emulsions with added hydrocolloids showed similar rheological patterns, whereas the control SPC exhibited a lower frequency dependence at the higher temperature (Figure 3b).

Table 3 shows the rheological property values derived from the frequency sweeps corresponding to the intermediate frequency of the applied range for the different formulated emulsions at both temperatures. At both temperatures, the type of cellulose ether incorporated into the emulsion had a significant (*p* < 0.05) effect on the viscoelastic properties of the emulsions, but the stage of CMC or MC addition to the emulsion also affected the values of some rheological properties. At 5 °C, SPC exhibited the highest *G*′ and *G*″ values and higher viscoelasticity (values of tan *δ* became closer to 0), revealing higher elasticity in the network structure of the emulsion in absence of hydrocolloids. The emulsions containing CMC had the lowest modulus values, and SPC/CMC–sim exhibited significantly lower viscoelasticity (higher tan *δ*) than SPC/CMC–seq. In turn, in the emulsions in which MC was added by simultaneous incorporation, *G*′ and *G*″ values significantly increased, whereas tan *δ* decreased, denoting a structural weakening in SPC/MC–seq compared with SPC/MC–sim at 5 °C. At 5 °C, lard is in a solid state, unlike at 37 °C when it is a liquid oil, which could justify the fact that oil droplets are crystallized and/or aggregated, with all emulsions showing very different rheological behavior.

In contrast, at 37 °C, the highest moduli values corresponded to emulsions containing both SPC and CMC, although the stage of hydrocolloid addition only had a significant effect on the storage modulus (*G*′) value, which was higher in SPC/CMC–seq than in SPC/CMC–sim (Table 3). At pH 7, both SPC and CMC were negatively charged [30]; therefore, as a consequence of the sequential addition of CMC in the emulsion, a gel-like network of small flocks was formed, which exhibited high viscoelastic moduli because flocculation by depletion was induced (Figure 1c), as previously observed in a whey protein (WP)-stabilized emulsion to which CMC was added using the sequential adsorption principle [15]. Consequently, CMC molecules stayed unabsorbed and formed a gel in the continuous phase of the emulsion with larger oil droplets locked in the network holes. As well, flocculation by depletion of emulsion droplets was also induced by adding CMC to emulsions that had been stabilized by *β*-lactoglobulin at pH 6.7 [36]. In addition, it was observed that the *G*′ value of emulsion SPC/CMC–sim was lower than that of SPC/CMC–seq, reflecting that a different mechanism of emulsion stabilization had occurred. In emulsion SPC/CMC–sim, SPC and CMC compete with each other for surface sites, SPC molecules being preferentially adsorbed due to their greater surface activity.

At 37 °C, there was no significant difference between the *G*′ value of the control emulsion SPC and that of the emulsions containing both SPC and MC (Table 3). However, emulsions SPC/MC–sim and SPC/MC–seq exhibited significantly higher *G*″ values than SPC, resulting in a significant decrease of viscoelasticity (tan *δ* values increased from 0.2 to 0.5). Interestingly, the stage of MC addition to the emulsion had no significant effect on the dynamic viscoelastic properties of emulsions SPC/MC at 37 °C and 1 Hz. The lack of significant differences between the *G*′ values of the control SPC, and emulsions SPC/MC–sim and SPC/MC–seq would indicate that in the mixed systems, SPC is adsorbed at the oil interface more rapidly, as it is more surface-active than MC, regardless of the emulsification stage in which MC was added. There are hardly any studies on the use of MC in the formulation of emulsions. In this context, the stability of a WP-stabilized emulsion in the presence of MC added after emulsification was previously investigated [19], reporting that because of the existence of a thermodynamic incompatibility between the negatively charged WP and the non-ionic MC, MC did not interact with WP.

#### 3.3.3. Temperature Sweep Tests

To simulate the effect of heating on the structure and to study the network stability of the emulsions during heating, temperature sweep tests were carried out from 5 to 60 °C. The variation of *G*′ and *G*″ with temperature for the five emulsions is shown in Figure 4, which exhibited differences in their heating patterns depending on the cellulose ether included in the emulsion. As depicted in Figure 4, compared with the control SPC, the addition of CMC or MC led to lower initial *G*′ values before and at the beginning of heating, but higher *G*′ values between 30 and 60 °C. Between 5 and 25 °C, in emulsions SPC, SPC/MC–sim, and SPC/MC–seq, the initial increase in temperature produced a rapid decrease in the values of both moduli, and with *G*′ values higher than those of *G*″. In addition, in sample SPC a crossover between *G*′ and *G*″ occurred at 25.6 ± 1.8 °C in response to the melting of the fat crystals, and above this temperature, SPC showed a liquid-like behavior (*G*″ values above *G*′). A crossover temperature was also detected during heating of both emulsions SPC/MC–sim and SPC/MC–seq (Figure 4), although the additional presence of MC in the emulsions exerted a delaying effect on this temperature. The crossover of *G*′ and *G*″ occurred at 37.8 ± 0.1 °C and at 46.6 ± 1.3 °C in emulsions SPC/MC–sim and SPC/MC–seq, respectively. This finding is associated with a significant increase in the melting temperature of the fat crystals in the emulsions with added MC, being much more evident when the sequential adsorption method was used. In addition, MC is known to show reversible thermal gelation. In emulsion SPC/MC–seq, *G*″ values were closer to those of *G*′ during all the heating process as compared with samples SPC and SPC/MC–sim, revealing lower viscoelasticity (higher tan *δ* values between 5 and 30 °C), but a solid-like mechanical behavior at 37 °C (tan *δ* < 1).

In contrast, the temperature-dependent behavior of the emulsions with added CMC was significantly different (Figure 4). The dominant elastic structure of both emulsions SPC/CMC–sim and SPC/CMC–seq was maintained throughout the complete temperature range analyzed (shown by *G*′ > *G*″). The behavior of the emulsions containing CMC was quite interesting. During heating, there was no crossover point and the emulsions retained their solid-like properties during the entire heating process, and both elasticity and viscosity only displayed a slight decrease with temperature, denoting a limited temperature effect between 5 and 60 °C without phase change.

The results suggest that the network structure of emulsions SPC/CMC was different from that of emulsions SPC/MC and that between 30 and 60 °C, CMC increased the rigidity of emulsion SPC more than MC. According to the supplier’s specifications, MC will gel reversibly at temperatures over 55 °C, whereas CMC does not exhibit reversible gel behavior.

### 3.4. Flow Behavior

The flow curves of the emulsions were obtained at 37 °C and fitted to both the power law model. The steady shear rheological parameters of the emulsions derived from the fits are shown in Table 4. All the emulsions displayed typical non-Newtonian behavior as the apparent viscosity decreased with the increase of shear rate (Appendix A). However, SPC/CMC–seq had the highest consistency coefficient (K) that correlated with the lowest flow behavior index (n) value, indicating that this emulsion had a more viscous consistency and higher pseudoplasticity due to a higher electrostatic interaction among both biopolymers. This result could be correlated with the lowest volume and average diameter values obtained for this emulsion (Table 2). A similar correlation between both power law parameters was observed in a fresh WP-stabilized O/W emulsion with added CMC using the sequential method [16,19], and the shear-thinning behavior was attributed to the structuring effect of CMC in the continuous phase of the emulsion, which impedes the flow of liquid through the network and reinforces bonds between oil droplets. SPC/CMC–sim also presented very high and low K and n values, respectively, although the results evidenced a significant loss of viscous behavior and pseudoplasticity in this emulsion in contrast with its SPC/CMC–seq counterpart. Similar results were obtained in the comparison of the simultaneous addition of WP and CMC and the successive addition of both biopolymers [15], reflecting that a dissimilar emulsion stabilization process occurs.

In turn, compared with SPC/CMC-stabilized emulsions, both emulsions stabilized by SPC/MC, and the one stabilized by SPC alone had significantly (*p* < 0.05) decreased K values, whereas n values increased (Table 4). This result was to be expected since, as it was mentioned in Section 2.1, the viscosity of CMC at 25 °C is also much higher than that of MC. However, there were no significant differences between the viscosity of SPC/MC–sim, SPC/MC–seq, and that of the control emulsion SPC with similar K values, but SPC/MC-stabilized emulsions exhibited significantly lower pseudoplasticity. The highest n values corresponded to SPC/MC–seq and SPC/MC–sim, in that order (Table 4), reflecting that the viscosity of these emulsions decreased more slowly with the increase of the shear rate than for the rest of the emulsions studied (Appendix A). In accordance with Pal [37], an increase in droplet size is accompanied by a decrease in the degree of shear-thinning behavior (rheological behavior closer to Newtonian) in concentrated emulsions, as observed in this study for SPC/MC-stabilized emulsions in comparison with those stabilized by SPC/CMC.

On the other hand, smaller particle sizes with lower D[4,3] and D[3,2] were found in SPC/CMC stabilized emulsions, which has been related to emulsion stabilization against creaming [38]. By reducing the droplet size, an increase in droplet volume occurs and, consequently, an increase in the interfacial area, which favors the increase in viscosity [39].

### 3.5. Three-Step Shear Rate Tests

A three-step shear rate analysis was conducted on the emulsions to test the amount of viscosity recovery following the breakdown of the emulsion’s structure (Appendix A). Table 5 shows the values of the parameters derived from the test for the emulsions. The original and final apparent viscosity (*η*_0_ and *η*_f_) values of emulsions were in consonance with the results derived from other rheological measurements. There were no significant differences between the percentages of viscosity recovery calculated with respect to the *η*_0_ value of the emulsions containing either SPC alone or SPC and CMC, whereas those corresponding to the emulsions stabilized by both SPC and MC were significantly lower. For the same recovery time of 6 min, emulsions SPC and SPC/MC–seq exhibited the fastest and the slowest rate of structure rebuilding, respectively. Therefore, all emulsions exhibited a time-dependent flow behavior as they did not completely recover their structure during the third stage of the test, but SPC/MC–seq displayed the highest time-dependent behavior, recovering only 35% of its viscosity. The recovery time of non-Newtonian fluids depends on both particle orientation and collision effects during structure rebuilding [25].

### 3.6. Texture Measurements

The textural parameter values of the different formulated emulsions derived from penetration tests are presented in Table 6. Both textural parameters were highly correlated, exhibiting the same significant differences between samples. At 5 °C, emulsions SPC/CMC–seq and SPC/MC–seq exhibited the highest and the lowest force at 10 mm and area under curve (AUC) values, indicating a higher and lower hardness or consistency in the network structure of these emulsions, respectively. This result would corroborate that flocculation by depletion could have been induced by using the sequential adsorption method in emulsions with added CMC, whereas the lowest gel consistency of emulsion SPC/MC–seq could be related to the higher capability of MC to be adsorbed at the interface of the SPC-stabilized emulsion. In turn, by using the simultaneous adsorption method, there were no significant differences between the textural parameters of emulsions containing SPC/CMC and SPC/MC, which reveals a similar gel consistency in these emulsions, regardless of the type of cellulose ether used. In addition, both emulsions SPC/CMC–sim and SPC/MC–sim showed lower values of force at 10 mm and AUC than the emulsion stabilized by SPC alone (Table 6), which reflects a weaker system at low temperatures, corroborating the results obtained from dynamic rheological tests (Table 3).

### 3.7. Extent of Lipolysis during In Vitro Digestion

The impact of the different formulations of the emulsions (SPC, SPC/CMC–sim, SPC/CMC–seq, SPC/MC–sim, and SPC/MC–seq) on the extent of lipolysis was evaluated by HPSEC analysis of the composition of the digestion products. The evaluation is based on the decrease of TAG as a result of the formation of hydrolytic compounds (DAG, and the sum of MAG and FFA), as previously described [40].

Table 7 shows the lipid profile of the different emulsions: initially (undigested pork lard), and after 30 min and 90 min of intestinal digestion. During the first 30 min, all emulsions presented a rapid release of hydrolytic products, as reflected by the considerable decrease in TAG concentration (below 50% in all samples). However, digestion between 30 and 90 min was lower, reaching a TAG concentration of 20–30%. Therefore, the velocity of lipolysis markedly decreased between 30 and 90 min of intestinal digestion. This fact suggests that regardless of the type of cellulose ether added and the stage of incorporation, lipase was adsorbed at the interface and hydrolyzed the lipids, resulting in the accumulation of lipolysis products (MAG and FFA) on the surface of the droplets, which, in turn, gradually slowed down lipase activity. This behavior was also observed by Malaki et al. [41], where the saturation of the O/W interface with bile salts, along with the accumulation of lipolytic products, such as FFA and MAG, inhibited any further access of pancreatin to its substrate. In accordance with Bellesi et al. [21], MAG and FFA tend to form a liquid crystalline phase that reduces the accessibility of lipase to the interface, thus preventing lipase from continuing to exert its activity.

As for the comparison among samples, after 30 min of intestinal digestion, the total percentage of lipid digestibility ranged between 58–63%, without significant differences among the emulsions. Although lipid digestibility and TAG and DAG levels were equivalent, significant differences in the absorbable fraction (MAG + FFA) can be observed. SPC/CMC-sim was the emulsion with the highest amount of MAG + FFA released, being significantly different from SPC/CMC-seq, which showed the lowest levels of absorbable fraction.

At the end of the in vitro digestion (90 min), the total percentage of lipid digestibility ranged between 64–77% and was significantly different among samples. All the emulsions stabilized by mixed biopolymers showed a significantly lower digestibility and a lower degree of lipolysis than emulsion SPC (lower FFA and MAG + FFA) (Table 7). This behavior was as expected, considering that emulsifier proteins such as soy (also whey and casein), which are usually employed in the formulation of food-grade emulsions, form an interfacial protein film that can be easily displaced by bile salts, facilitating lipid digestion by lipases. This is consistent with the thin film of SPC surrounding the fat droplets in the control emulsion SPC (Figure 2a).

On the other hand, emulsions formulated by sequential adsorption (SPC/CMC–seq and SPC/MC–seq) exhibited the highest TAG content and the lowest MAG + FFA content and degree of lipolysis; while the emulsions made by the co-adsorption method (SPC/CMC–sim and SPC/MC–sim), although showing significantly less digestion that the control SPC, seemed slightly less efficient in blocking lipid digestion than their sequential counterparts. Comparing SPC/CMC–seq with SPC/CMC–sim, only FFA levels were significantly lower in the emulsion made by the sequential adsorption method. However, the differences between SPC/MC–seq and SPC/MC–sim were much more evident regarding TAG, MAG, FFA, MAG + FFA contents, and percentage digestibility.

Globally, the emulsion made by co-adsorption showed more digestion, and this could be related to a disassociation between the protein and the biopolymer (SPC and CMC or MC) during stomach digestion, caused by pepsin-driven proteolysis of the interfacial protein. In contrast, in the emulsions prepared by sequential adsorption, when the cellulose ethers were added after SPC adsorption, they were not affected by pepsin during the gastric phase and, consequently, the interfacial film and emulsions containing MC or CMC and SPC had a more intact structure once exposed to duodenal conditions. Overall, the sequential method, compared with the simultaneous method, results in a slower rate and lower extent of FFA release. Furthermore, these results can be also attributed to the thick interfacial film observed around the oil droplets in the emulsions formulated using the sequential adsorption method (Figure 2b,c).

In addition to the effect of the stage of polysaccharide incorporation (sequential or simultaneous), the type of cellulose ether (CMC or MC) was also analyzed. In this regard, there were no significant differences between SPC/CMC-sim and SPC/MC-sim (Table 7); SPC/MC-seq exhibited the highest TAG levels at the end of intestinal digestion (36%), as well as the lowest content of absorbable fraction (48.6%), being significantly different from the other emulsions tested. Such differences found in the polysaccharides when sequentially added to the emulsion could be related to the fact that MC is hydrophobic and non-ionic, conferring more affinity between the oil/water interface than an-ionic CMC, which limits the access of lipase to the oil droplets. Likewise, Torcello-Gómez and Foster [42] reported the different binding of bile salts to different types of cellulose ethers, postulating the existence of hydrophobic interactions between these components depending on the cellulose derivative. Pizones Ruiz-Henestrosa et al. [43] also demonstrated the binding between bile salts and two non-ionic cellulose ethers with different molecular structures, both in the aqueous phase and O/W interface.

Given that pork fat is one of the ingredients commonly used in the formulation of meat products and its reduction/elimination causes negative effects on sensory properties, the incorporation of these emulsions in products type pâté, sausages, etc., offers important possibilities as a potential strategy, to reduce the fat content and its absorption.

## 4. Conclusions

From the results obtained, it can be concluded that the type and the stage of incorporation of CMC and MC during the preparation of emulsions significantly modify the structural and rheological properties of the systems as well as the in vitro lipid digestion. During the first 30 min, all emulsions presented a rapid release of hydrolytic products and the velocity of lipolysis markedly decreased between 30 and 90 min of intestinal digestion. All emulsions stabilized by mixed biopolymers showed a significantly lower digestibility and lower degree of lipolysis than emulsion SPC. Emulsions formulated by sequential adsorption presented a lower degree of lipolysis than emulsions made by the co-adsorption method. SPC/MC-seq exhibited the highest TAG levels at the end of intestinal digestion (36%), as well as the lowest content of absorbable fraction (48.6%) likely due to the fact that MC is hydrophobic and non-ionic, conferring more affinity between the oil/water interface than an-ionic CMC, which limits the access of lipase to the oil droplets. Indeed, all the distinctive rheological features observed in emulsion SPC/MC–seq at 37 °C (larger LVE range and greater structural stability, decreased viscoelasticity and pseudoplasticity, and higher thixotropic behavior) could be related to its lower lipid digestibility.

## Figures and Tables

**Figure 1 foods-11-00738-f001:**
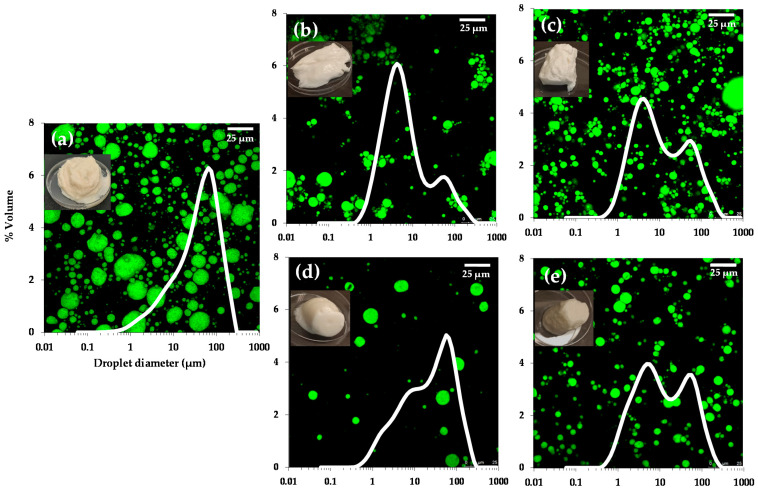
Confocal micrographs of SPC-stabilized emulsions containing CMC and MC added at two different stages of the emulsion formulation. (**a**) SPC; (**b**) SPC/CMC–sim; (**c**) SPC/CMC–seq; (**d**) SPC/MC–sim; (**e**) SPC/MC–seq. Scale bar = 25 µm. General appearance and particle-size distributions of the emulsions are superimposed on the micrographs.

**Figure 2 foods-11-00738-f002:**
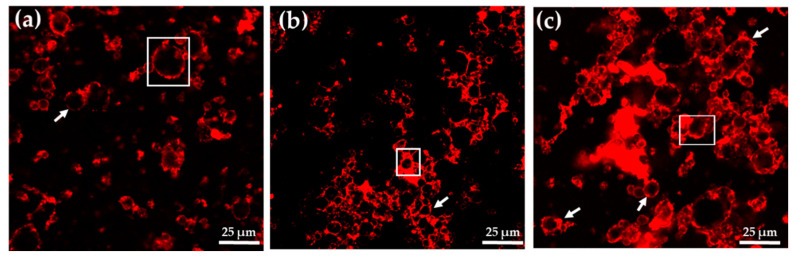
Confocal micrographs of SPC-stabilized emulsions containing CMC and MC added by sequential adsorption. (**a**) SPC; (**b**) SPC/CMC–seq; (**c**) SPC/MC–seq. Scale bar = 25 µm. Boxes and arrows indicate a thin protein layer in emulsion SPC and a thicker one in emulsions with CMC and MC.

**Figure 3 foods-11-00738-f003:**
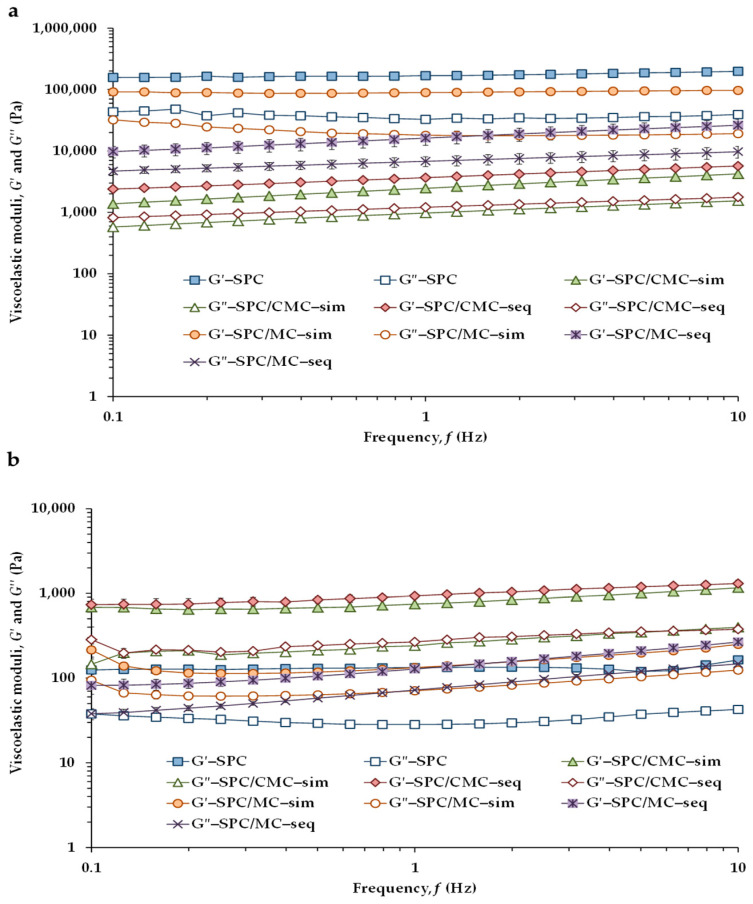
Viscoelastic moduli (*G*′: filled symbols; *G*″: open symbols) as a function of the frequency for the emulsions: (**a**) at 5 °C; (**b**) at 37 °C.

**Figure 4 foods-11-00738-f004:**
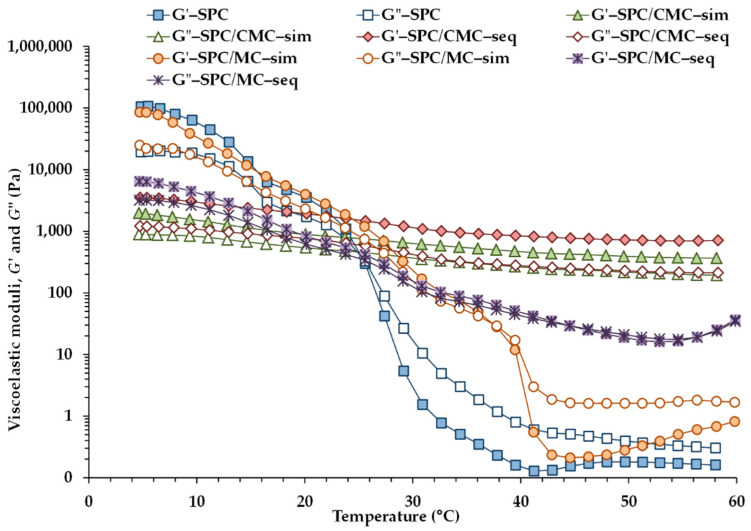
Viscoelastic moduli (*G*′: filled symbols; *G*″: open symbols) as a function of temperature between 5 and 60 °C for the emulsions.

**Table 1 foods-11-00738-t001:** Fatty acid composition of the pork lard and emulsions formulated.

mg Fatty Acid/g Sample	Pork Lard	Emulsions *
Palmitic C16:0	245 ± 2.0 ^a^	96.0 ± 0.23 ^b^
Estearic C18:0	154 ± 0.70 ^a^	50.9 ± 0.16 ^b^
Other SFA	18.7	8.10
∑SFA	419	155
Oleic C18:1n9	408 ± 5.2 ^a^	162 ± 0.52 ^b^
Other MUFA	49.7	23.6
∑MUFA	458	186
Linoleic C18:2n6	65.4 ± 0.71 ^a^	28.1 ± 0.081 ^b^
Other PUFA	13.3 ± 0.14 ^a^	5.60 ± 0.046 ^b^
∑PUFA	78.8	33.7
∑PUFA/∑SFA	0.188	0.217

^a,b^ Values followed by the same letter within each row indicate no significant differences (*p* < 0.05). For pork lard, mean value (*n* = 3) ± standard deviation. * Since all the emulsions contain the same type and amount of fat, the values given correspond to the average of the five formulated emulsions.

**Table 2 foods-11-00738-t002:** Particle size diameters for the different formulated emulsions.

Emulsion	D[4,3] (μm)	D[3,2] (μm)
SPC	52.9 ± 1.1 ^a^	11.6 ± 0.43 ^a^
SPC/CMC–sim	23.8 ± 0.65 ^d^	4.23 ± 0.035 ^c^
SPC/CMC–seq	15.7 ± 0.35 ^e^	3.46 ± 0.12 ^d^
SPC/MC–sim	38.1 ± 2.0 ^b^	6.74 ± 0.10 ^b^
SPC/MC–seq	26.8 ± 0.17 ^c^	4.50 ± 0.029 ^c^

Mean value (*n* = 3) ± standard deviation. D[4,3]: volume average diameter; D[3,2]: surface average diameter. ^a–e^ Values followed by the same letter within each column indicate no significant differences (*p* < 0.05).

**Table 3 foods-11-00738-t003:** Viscoelastic properties at 1 Hz obtained at 5 and at 37 °C from frequency sweep tests, for the different formulated emulsions.

Emulsion	*G*′ (Pa)	*G*″ (Pa)	tan *δ*
**At 5 °C**			
SPC	168,300 ± 42,500 ^a^	32,670 ± 3240 ^a^	0.199 ± 0.032 ^c^
SPC/CMC–sim	2468 ± 59 ^c^	974 ± 28 ^d^	0.393 ± 0.0020 ^a^
SPC/CMC–seq	3671 ± 227 ^c^	1211 ± 54 ^d^	0.330 ± 0.0058 ^b^
SPC/MC–sim	89,260 ± 270 ^b^	17,995 ± 405 ^b^	0.202 ± 0.0051 ^c^
SPC/MC–seq	16,190 ± 1886 ^c^	6789 ± 337 ^c^	0.426 ± 0.029 ^a^
**At 37 °C**			
SPC	133 ± 16 ^c^	28.3 ± 0.0035 ^c^	0.214 ± 0.0075 ^c^
SPC/CMC–sim	745 ± 31 ^b^	240 ± 13 ^a^	0.322 ± 0.0047 ^b^
SPC/CMC–seq	948 ± 75 ^a^	268 ± 0.015 ^a^	0.285 ± 0.038 ^b^
SPC/MC–sim	133 ± 0.20 ^c^	70.9 ± 1.2 ^b^	0.533 ± 0.0079 ^a^
SPC/MC–seq	128 ± 11 ^c^	72. ± 3.3 ^b^	0.566 ± 0.023 ^a^

Mean value (*n* = 3) ± standard deviation. *G*′, storage modulus; *G*″, loss modulus; tan *δ*, loss factor (=*G*″/*G*′). ^a–d^ For the same temperature, values followed by the same letter within each column indicate no significant differences (*p* < 0.05).

**Table 4 foods-11-00738-t004:** Steady shear rheological parameters at 37 °C derived from fits to the power law model for the different formulated emulsions.

Emulsion	*K* (Pa s^n^)	*n* (-)	*R* ^2^
SPC	1.12 ± 0.058 ^c^	0.360 ± 0.0070 ^d^	0.967 ± 0.0061
SPC/CMC–sim	64.8 ± 1.1 ^b^	0.390 ± 0.00035 ^c^	0.998 ± 0.000050
SPC/CMC–seq	81.9 ± 4.9 ^a^	0.226 ± 0.0056 ^e^	0.998 ± 0.00080
SPC/MC–sim	2.35 ± 0.13 ^c^	0.450 ± 0.016 ^b^	0.963 ± 0.0021
SPC/MC–seq	2.04 ± 0.10 ^c^	0.565 ± 0.012 ^a^	0.990 ± 0.0045

Mean value (*n* = 3) ± standard deviation. *K*, consistency coefficient; *n*, flow behavior index; *R*^2^, coefficient of determination. ^a–e^ Values followed by the same letter within each column indicate no significant differences (*p* < 0.05).

**Table 5 foods-11-00738-t005:** Steady shear rheological parameters at 37 °C derived from three-step shear rate tests for the different formulated emulsions.

	*η*_o_ (Pa s)	*η*_f_ (Pa s)	Viscosity Recovery (%)
SPC	5.04 ± 0.35 ^c^	3.49 ± 0.41 ^d^	69.0 ± 3.5 ^a^
SPC/CMC–sim	645 ± 24 ^b^	403 ± 8.1 ^b^	64.6 ± 11 ^a^
SPC/CMC–seq	1852 ± 128 ^a^	1209 ± 33 ^a^	65.7 ± 6.3 ^a^
SPC/MC–sim	49.9 ± 1.1 ^c^	20.5 ± 0.21 ^c,d^	41.0 ± 0.51 ^b^
SPC/MC–seq	168 ± 5.9 ^c^	58.8 ± 0.61 ^c^	35.1 ± 0.86 ^b^

Mean value (*n* = 3) ± standard deviation. *η*_o_, original apparent viscosity; *η*_f_, final apparent viscosity; *η*_f_ × 100/*η*_o_, percentage of viscosity recovery at the end of the test. ^a–d^ Values followed by the same letter within each column indicate no significant differences (*p* < 0.05).

**Table 6 foods-11-00738-t006:** Textural penetration parameters at 5 °C for the different formulated emulsions.

Emulsion	Force at 10 mm (N)	AUC (N s)
SPC	0.689 ± 0.042 ^b^	8.31 ± 0.62 ^b^
SPC/CMC–sim	0.462 ± 0.014 ^c^	5.52 ± 0.15 ^c^
SPC/CMC–seq	0.789 ± 0.033 ^a^	9.76 ± 0. 50 ^a^
SPC/MC–sim	0.427 ± 0.0085 ^c^	5.23 ± 0.25 ^c^
SPC/MC–seq	0.246 ± 0.0050 ^d^	3.00 ± 0.055 ^d^

Mean value (*n* = 3) ± standard deviation. AUC, area under curve. ^a–d^ Values followed by the same letter within each column indicate no significant differences (*p* < 0.05).

**Table 7 foods-11-00738-t007:** Fat composition and lipid digestibility of emulsions during in vitro digestion.

Emulsion	TAG (g/100 g)	DAG (g/100 g)	MAG (g/100 g)	FFA (g/100 g)	MAG + FFA (g/100 g)	Digestibility (%)
Initial pork lard	99.9 ± 0.0					-
**Digestion time: 30 min**						
SPC	41.2 ± 2.5 ^a^	14.9 ± 1.8 ^a^	11.9 ± 0.90 ^a^	32.0 ± 0.50 ^a^	43.9 ± 2.5 ^a,b^	58.8 ± 2.5 ^a^
SPC/CMC–sim	36.6 ± 2.0 ^a^	16.6 ± 1.2 ^a^	12.1 ± 0.53 ^a^	34.8 ± 2.3 ^a^	46.9 ± 2.0 ^a^	63.4 ± 2.1 ^a^
SPC/CMC–seq	39.5 ± 3.2 ^a^	15.6 ± 0.75 ^a^	11.5 ± 0.68 ^a,b^	33.4 ± 1.8 ^a^	44.9 ± 3.2 ^a,b^	60.5 ± 3.2 ^a^
SPC/MC–sim	39.0 ± 0.21 ^a^	16.7 ± 0.14 ^a^	12.7 ± 0.72 ^a^	31.6 ± 1.1 ^a^	44.2 ± 0.21 ^a,b^	61.0 ± 0.21 ^a^
SPC/MC–seq	41.2 ± 0.26 ^a^	17.7 ± 0.10 ^a^	9.77 ± 0.37 ^b^	31.4 ± 0.73 ^a^	41.1 ± 0.26 ^b^	58.8 ± 0.26 ^a^
**Digestion time: 90 min**						
SPC	22.7 ± 0.53 ^d^	13.5 ± 2.0 ^a^	15.1 ± 0.69 ^a^	50.2 ± 0.69 ^a^	65.3 ± 0.00 ^a^	77.3 ± 0.53 ^a^
SPC/CMC–sim	27.0 ± 0.69 ^b,c^	14.7 ± 0.39 ^a^	13.3 ± 0.90 ^a,b^	45.0 ± 0.77 ^b,c^	58.3 ± 0.87 ^b^	73.0 ± 0.69 ^b,c^
SPC/CMC–seq	30.9 ± 1.6 ^b^	15.1 ± 0.38 ^a^	11.8 ± 0.71 ^b,c^	42.3 ± 1.2 ^c^	54.1 ± 1.9 ^c^	69.2 ± 1.6 ^c,d^
SPC/MC–sim	25.9 ± 1.2 ^c,d^	13.2 ± 1.5 ^a^	14.8 ± 1.1 ^a^	46.2 ± 1.5 ^b^	61.0 ± 0.46 ^b^	74.1 ± 1.2 ^b^
SPC/MC–seq	36.3 ± 2.4 ^a^	15.1 ± 1.1 ^a^	10.2 ± 1.3 ^c^	38.4 ± 1.2 ^d^	48.6 ± 1.8 ^d^	63.7 ± 2.4 ^d^

Mean value (*n* = 3) ± standard deviation. TAG: triacylglycerides; DAG: diacylglycerides; MAG: monoacylglyceride; FFA: free fatty acids. ^a–d^ For the same digestion time, values followed by the same letter within each column indicate no significant differences (*p* < 0.05).

## Data Availability

Data is contained within the article or Appendix A.

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
