# Peer review of "The Influence of Cellulose Ethers on the Physico-Chemical Properties, Structure and Lipid Digestibility of Animal Fat Emulsions Stabilized by Soy Protein"

_foods, 2022, doi:10.3390/foods11050738_

Round 1

Reviewer 1 Report

This study investigated that the influence of cellulose ethers and its addition order on the physicochemical properties, structure and lipid digestibility of animal fat emulsions stabilized by soy protein. Overall, a lot of experiments were designed purposefully and conducted systematically. Accordingly, all the results are shown in detail and analyzed carefully. It is a hard, interesting work. The obtained results and conclusion will contribute to the development of healthy fat and functional meat products.

Specific comments are listed below.

1 In line 207, it may better if “samples”is replaced with “points”. After all, the latter is used more commonly.

2 In line 298, the formula mentioned is absent.

3 In line 333, “aspectwith” should be “aspect with”. Similar flaws exist otherwhere, such as that in 365.

4 In Fig. 1, The fat content seems not to be same among all groups with respect to the number and size of the oil drops; after adding cellulose ether, the fat content in emulsions reduced, especially, as shown in in (c) and (d).

5 Maybe, it is difficult to obtain the information and related hypothesis (in line 371-377) from the images shown in Figure 2. Additionally, it is needed to check which type of cellulose ether (CMC or MC) on earth shown in (b).

Author Response

Comments and Suggestions for Authors

This study investigated that the influence of cellulose ethers and its addition order on the physicochemical properties, structure and lipid digestibility of animal fat emulsions stabilized by soy protein. Overall, a lot of experiments were designed purposefully and conducted systematically. Accordingly, all the results are shown in detail and analyzed carefully. It is a hard, interesting work. The obtained results and conclusion will contribute to the development of healthy fat and functional meat products.

Thank you very much for your kind comments related to our manuscript.

Specific comments are listed below.

1 In line 207, it may better if “samples”is replaced with “points”. After all, the latter is used more commonly.

It was changed as suggested.

2 In line 298, the formula mentioned is absent.

Thank you very much for this comments. Honestly, formula for calculating digestibility of samples was not included due to an oversight. It has been added in the revised version.

3 In line 333, “aspectwith” should be “aspect with”. Similar flaws exist otherwhere, such as that in 365.

These flaws have been corrected, here and elsewhere.

4 In Fig. 1, The fat content seems not to be same among all groups with respect to the number and size of the oil drops; after adding cellulose ether, the fat content in emulsions reduced, especially, as shown in in (c) and (d).

For the analysis of the microstructure of the emulsions, we take several photographs of the same preparation in different parts to try to see the structure in a global way. However, this is sometimes not easy. Regarding the fat content, although similar in all the emulsions, it is not easy to compare since the particle size and distribution is different in each type of emulsion (Table 2) as a consequence of the different composition, which affects its microstructure. In the photograph corresponding to sample SPC-CMC-seq (Figure 1c) the amount of fat would be similar to the rest but as discussed throughout the manuscript, there is the phenomenon of depletion flocculation and the fat droplets are not distributed in the same way as in the rest. On the other hand, perhaps, the photograph corresponding to sample SPC/MC-sim (Figure 1d) was taken in a margin of the preparation, so we have looked for another one that is more representative and we have replaced it.

5 Maybe, it is difficult to obtain the information and related hypothesis (in line 371-377) from the images shown in Figure 2. Additionally, it is needed to check which type of cellulose ether (CMC or MC) on earth shown in (b).

The Fast Green chromophore stains proteins and polysaccharides and with the conditions applied in the confocal microscopy assay, they are visualized in red surrounding the fat droplets that appear in black. What the authors appreciate in these photographs is that in the emulsion stabilized only with SPC (Figure 2a) a thin layer of protein appears surrounding the fat droplets (indicated in the boxes and white arrows), while in the emulsions stabilized with the mixture of SPC -CMC and SPC-MC the layer that surrounds the drop of fat is thicker, and this effect is more appreciated in the emulsion with MC.

For clarification, a new Figure 2 has been included.On the other hand, in legend to Figure 2 is indicated that confocal micrographs (b) and (c) correspond to SPC-stabilized emulsions containing CMC and MC added by sequential adsorption, respectively.

Reviewer 2 Report

Thank you for an interesting and well written manuscript on a topical aspect of the food industry. The introduction and materials and methods sections were appropriate.  

Is it possible to expand on your discussion with regards to the mechanism by which the cellulose ethers affected the particle size of the materials ? Could you look for recent papers (2021 and 2022) to explain this in more detail?

Was the rheological data significantly correlatable to the particle size and also the molecular structure of the mixtures ?

I am a bit surprised that temperature did not affect the rheology very much. What are your thoughts about why temperature was not a factor. 

The fat composition and digestion information was interesting. How could this be applied to real food products ?

Author Response

Thank you for an interesting and well written manuscript on a topical aspect of the food industry. The introduction and materials and methods sections were appropriate.  

Thank you very much for your kind comments regarding our manuscript.

Is it possible to expand on your discussion with regards to the mechanism by which the cellulose ethers affected the particle size of the materials? Could you look for recent papers (2021 and 2022) to explain this in more detail?

Most recent articles on this topic evaluate the effect of different cellulose ethers as emulsifiers for vegetable oils, however, there are hardly any studies that analyze the behavior of these hydrocolloids on emulsions stabilized with emulsifying proteins such as soy protein and less made with pork lard. For this reason, it is difficult to compare behaviors. Nevertheless, a sentence has been include concerning the indicated mechanism.

Was the rheological data significantly correlatable to the particle size and also the molecular structure of the mixtures?

Yes, mainly the rheological data from flow behavior (K and n values) are correlated with the particle size of the different emulsions. The following sentence has been included under 3.4. Flow Behavior subsection “This result could be correlated with the lowest volume and average diameter values obtained for this emulsion (Table 2)” in order to emphasize this aspect. On the other hand, it was mentioned in the original version that “In accordance with Pal [37], an increase in droplet size is accompanied by a decrease in the degree of shear-thinning behavior (rheological behavior closer to Newtonian) in concentrated emulsions, as observed in this study for SPC/MC-stabilized emulsions in comparison with those stabilized by SPC/CMC”.

I am a bit surprised that temperature did not affect the rheology very much. What are your thoughts about why temperature was not a factor. 

Well, from our point of view and from the results presented in both Table 3 and Figure 4, it is possible to observe that temperature affected quite the rheological behavior of the emulsions. The viscoelastic behavior of all the emulsions was very different at 5 and at 37 °C (Table 3). At 5 °C, SPC exhibited the highest G′ and G′′ values and higher viscoelasticity, whereas at 37 °C the highest moduli values corresponded to emulsions containing both SPC and CMC. On the other hand, Figure 4 evidences that under heating between 5 and 60 °C, the emulsions exhibited different heating pattern depending on the cellulose ether added.

We agree with the reviewer, and certainly, temperature affected less emulsions SPC/CMC–sim and SPC/CMC–seq because there was no crossover point of G' and G", and the emulsions retained their solid-like properties during the entire heating process. In this sense, on the line 523, the statement “denoting a limited structural breakdown between 5 and 60 °C” has been changed to “denoting a limited temperature effect between 5 and 60 °C without phase change”.  

The different behavior against temperature exhibited by emulsions SPC/CMC–sim and SPC/CMC–seq in comparison to emulsions SPC/MC–sim and SPC/MC–seq, could be ascribed to that CMC has much higher viscosity than MC (30,000 mPa s in 2% solution at 25 °C vs. 4,000 mPa s at 2% in water at 25 °C). In addition, according to supplier's specifications, MC will gel reversibly gel at temperatures over 55 °C, whereas CMC CMC does not exhibit a behavior of reversible gel. The latter has been added in the text of the revised version.

The fat composition and digestion information was interesting. How could this be applied to real food products ?

A sentence about the possible practical applications of this type of emulsions to real foods, more concretely meat products, has been included before the conclusions.

Reviewer 3 Report

In this work, the authors investigated the effects f using different cellulose ethers (anionic CMC and non-ionic MC) on the physicochemical and structural properties, as well as on the lipid digestibility of pork lard O/W emulsions stabilized by soy protein concentrate. The work is well-written, well-organized, and is of interest to the food industry and to the readers.

1. The English language can be improved.

2. Based on the fat content (40%), and the appearance of the system (Figure 1), do you think the system is an emulsion or gel-like emulsion (emulsion gel), kindly change the name if needed. 

3. More in-depth discussions are recommended in the results and discussions section to improve the quality of this article. 

Author Response

In this work, the authors investigated the effects f using different cellulose ethers (anionic CMC and non-ionic MC) on the physicochemical and structural properties, as well as on the lipid digestibility of pork lard O/W emulsions stabilized by soy protein concentrate. The work is well-written, well-organized, and is of interest to the food industry and to the readers.

Thank you very much for your kind comments regarding our manuscript.

  1. The English language can be improved.

Regarding to the improvement of English language suggested, we would like to say that the original manuscript was checked by a native English-speaking colleague before sending it to the Journal.

  1. Based on the fat content (40%), and the appearance of the system (Figure 1), do you think the system is an emulsion or gel-like emulsion (emulsion gel), kindly change the name if needed. 

Well, certainly, from a rheological point of view, and from the appearance of the systems shown in the Figure 1, maybe it would seem more correct to speak in terms of gel-like emulsions. As mentioned in the text, at 5 °C all emulsions presented a higher storage modulus (G′) than the loss modulus (G′′), and in the frequency range studied, both moduli showed a considerable frequency dependence, therefore exhibiting a weak gel-like or structured liquid behavior. However, as it was indicated in 3.3.3. Temperature Sweep Tests subsection, the control SPC-stabilized emulsion shows a liquid-like behavior (with G" values above G') at 37 °C, which is the temperature used in in vitro digestions. Therefore, it is expected that control SPC emulsion behaves like a liquid during digestion. For this reason, we would prefer to name emulsions to the systems formulated better than emulsion gel. However, we are aware that the structural reinforcement observed in the rest of the emulsions is clearly due to the presence of cellulose ethers.

  1. More in-depth discussions are recommended in the results and discussions section to improve the quality of this article. 

As the reviewer comments, the work is well-written, well-organized, and is of interest to the food industry and to the readers. In addition, another reviewer tells us that, all the results are shown in detail and analyzed carefully. As such, the manuscript is already perhaps too long. And therefore, we would prefer not to include further discussion.